# A historical evaluation of the disease avoidance theory of xenophobia

Ceyhun Sunsay *

Kutztown University of Pennsylvania, Kutztown, PA, United States of America

* sunsay@kutztown.edu

**Data Availability Statement:** The data, the corpus and the program codes underlying the results presented in the study are available from the Open Science Foundation via https://osf.io/4f7ty/.

**Funding:** The authors received no specific funding for this work.

## Abstract

Historical psychology is emerging as a multidisciplinary field for studying psychological phenomena in a historical context. Historical records can also serve as testbeds for psychological theories, particularly the evolutionary ones. In Study 1 we aimed to gather evidence to evaluate the disease avoidance theory of xenophobia by analyzing the narratives of European explorers from the 15th and 16th centuries. Contrary to the theory's expectations, the narratives revealed numerous instances of close physical contact between the explorers and the native populations. Furthermore, rather than using disgust-laden words, the explorers portrayed the natives in a positive light. In Study 2, we employed a word embedding algorithm to explore whether native group names and their unfamiliar appearance were associated with disgust-laden words in the 19th century travel literature. The results indicated that while native group names showed such associations, their appearance did not. Finally, through network analysis, we demonstrated that embedded words such as "savages" mediated the perception of native groups as potential disease-threat. The findings highlight the significance of cultural factors that evolve over time, rather than cognitive adaptations believed to have evolved prior to the emergence of human culture, in explaining xenophobia.

## Introduction

History and psychology are strange bed fellows. History deals with past events determined by social and economic causes, while psychology deals with current behaviors and cognitions determined by individual motivations. Yet some scholars have long thought that bringing these two disciplines together may serve as a nuptial bed to beget new ideas. At the turn of the 20th century, a school of French historians (i.e., the Annales school) began to integrate psychological findings and perspectives into their research and explanations to gain a more comprehensive understanding of historical events. Some historians such as Karl Lamprecht in 1905 even called their discipline social psychological science [1]. Likewise, in the other camp some psychologists such as Ignace Meyerson called for studies on the historicity of the human mind and social psychology [2, 3]. More recent conceptual rapprochements agree that historical context is a crucial determinant of social psychological phenomena and consequently social psychology studies should not be limited with the present social-historical context [4–10]

**Competing interests:** The authors have declared that no competing interests exist.

Rapprochement between these fields provides unique opportunities for testing social psychological theories as well; if universal motivations are the root causes of psychological manifestations as we generally assume [11], we should expect a temporal continuity in human motivations in explaining human behaviors across the centuries. Recent calls have highlighted the importance of testing psychological theories with historical record of past behaviors and cognitions [4, 6, 8]. There have been a few studies that utilized historical data from relatively recent decades to assess psychological phenomena and theories [12, 13]. Historical data spanning centuries has been utilized to study the effects of economic and political conditions on aggression, creativity, leadership, and personality [14, 15]. Historical data may especially be useful to evaluate evolutionary theories in social psychology. Evolutionary accounts posit that variables that affected the lives of human ancestors during the Pleistocene age or earlier were ingrained into the mind as psychological adaptations that continue to influence our decisions even today [16]. Historical data might help us assess whether such primeval adaptations are historically invariant as expected by the evolutionary theories and the extent of human culture shaping these adaptations across the centuries [6]. Yet only a number of studies were conducted to assess whether purported evolutionary psychological adaptations are relatively time invariant or subject to cultural influence. These studies used a cross-generational method to compare behavioral patterns in mate preference across several decades in the 20[th] century [17–19]. Testing historical continuity of human motivations across longer spans of time is largely missing in the literature. One of our objectives is to show how historical data may help us assess the extent of historical cultural context in shaping our psychological architecture by using the disease avoidance adaptation as a case in point.

## Study 1

One of the primeval psychological adaptations posited to exist, protects organisms from contagious diseases. Disease is a serious expenditure of energy that can otherwise be utilized for the procurement of resources such as foods and mates as well as a threat to the survival of individuals. The disease avoidance adaptation or the behavioral immune system evolved as a first line of defense to protect organisms from contagion by arousing disgust and initiating avoidance behaviors [20, 21]. Because pathogens are invisible, the system is triggered by the perception of disease symptoms in addition to biological wastes. However, inferring the presence of disease from symptoms is an error-prone process since not every symptom signifies the presence of pathogens. The system is thus evolutionarily set to "interpret" innocuous symptoms such as acnes or blemishes as disease symptoms as well, since making false positive errors reduces the chance of infection and increases the safety of organisms.

The theory makes further predictions that are of interest to the readers of social psychology. Since the disease avoidance system is liberal to make false positive errors, it also "interprets" non-standard (i.e., unfamiliar) characteristics such as racial morphological features as potential disease symptoms. This bias according to the theory contributes to xenophobic tendencies. Indeed, there is a substantial number of correlational and a few experimental results supporting the theory [22–24]

Because evolutionary adaptations predate ancient civilizations, the purported relationship between xenophobia and the disease avoidance adaptation should be a recurring theme in the historiography of civilizations. History as an immense database of human psychological manifestations could be mined to test the validity of the disease avoidance theory of xenophobia across the history. The advocates of the theory have presented anecdotal historical evidence for the association between foreigners and disease in their works [24, 25]. We propose that a historical investigation of the relationship between foreignness and disease avoidance should assess the theory with objective data rather than anecdotal ones.

The first centuries of the era known as the age of exploration when Europeans made the first contacts with hitherto unknown people is of particular importance. The theory would anticipate that the indigenous people and their morphological characteristics should be associated with words connoting disease and disgust in the narratives of the explorers. Furthermore, the theory anticipates plenty of avoidance behaviors and fewer physical contacts with indigenous people in these narratives.

## Method

**Selection of resources.**   The age of exploration spans several centuries starting in the first quarter of the 15$^{th}$ century when Henry the navigator sent expeditions to Western Africa, ending in the 18$^{th}$ century with the discovery of Hawaiian archipelago. Examining historical records of such a long era is impractical and fraught with many problems. Economic and cultural changes that took place over the centuries may potentially confound the interpretation of the data. We chose the narratives of explorers who made the first contact with indigenous inhabitants in Western Africa and North and South America in the 15$^{th}$ and 16$^{th}$ centuries. As early as the 1520's Europeans had already started trading with various native American tribes [26]. Consequently, native North American populations had already established firm acquaintance with the Europeans by the middle of the 16$^{th}$ century. We therefore restricted our focus to the period between the 1450's and the 1550's when the unfamiliarity between European explorers and native populations was the highest. The list of explorers whose narratives fit our criteria were Andagoya, Azurara, Cabeza de Vaca, Cabral, Cadamosto, Cartier, Columbus, De Castenada, De Soto, Drake, Gaspar, Magellan, Vasco de Gama, Verrazano, and Vespucci. Some of these explorers published several narratives giving us a total of eighteen narratives.

Units of Analysis. We read the sections of the narratives where interactions between native inhabitants and Europeans occurred and calculated the term frequency (TF) of the target words of contact and avoidance as well as disgust and pleasant words in each narrative. We filtered the behaviors and words that specifically described physical contact or avoidance of contact between the Europeans and indigenous people in each narrative, excluding those that referred to situations or objects (e.g., "we avoided the hurricane" or "the island has beautiful trees").

## Results

Disease terms were relatively common in the corpus (N = 194) indicating an acute presence of diseases. Despite the salient threat of diseases, avoidance of physical contact between the groups (i.e., explorers and native people) was scarce. In contrast, there were many physical contacts between the groups (see Fig 1). A chi-square analysis tested whether total number of behaviors distributed equally between the contact and avoidance categories. The analysis showed more contact behaviors than avoidance behaviors, $\chi^2(45) = 41.09$, $p<0.001$. A similar analysis showed that the frequency of the words connoting disgust was significantly smaller in comparison to the frequency of near-antonyms of disgust-connoting words (see Fig 2). A chi-square analysis confirmed this pattern, $\chi^2(36) = 9.0$, $p<0.0027$. This pattern was observed in most of the narratives (see Appendix A in S1 File).

## Discussion

The narratives indicated an acute presence of disease which is expected to heighten disease avoidance tendencies, according to the disease avoidance theory [21]. However, explorers did not report disgust aroused by the unfamiliar morphological characteristics of native people or

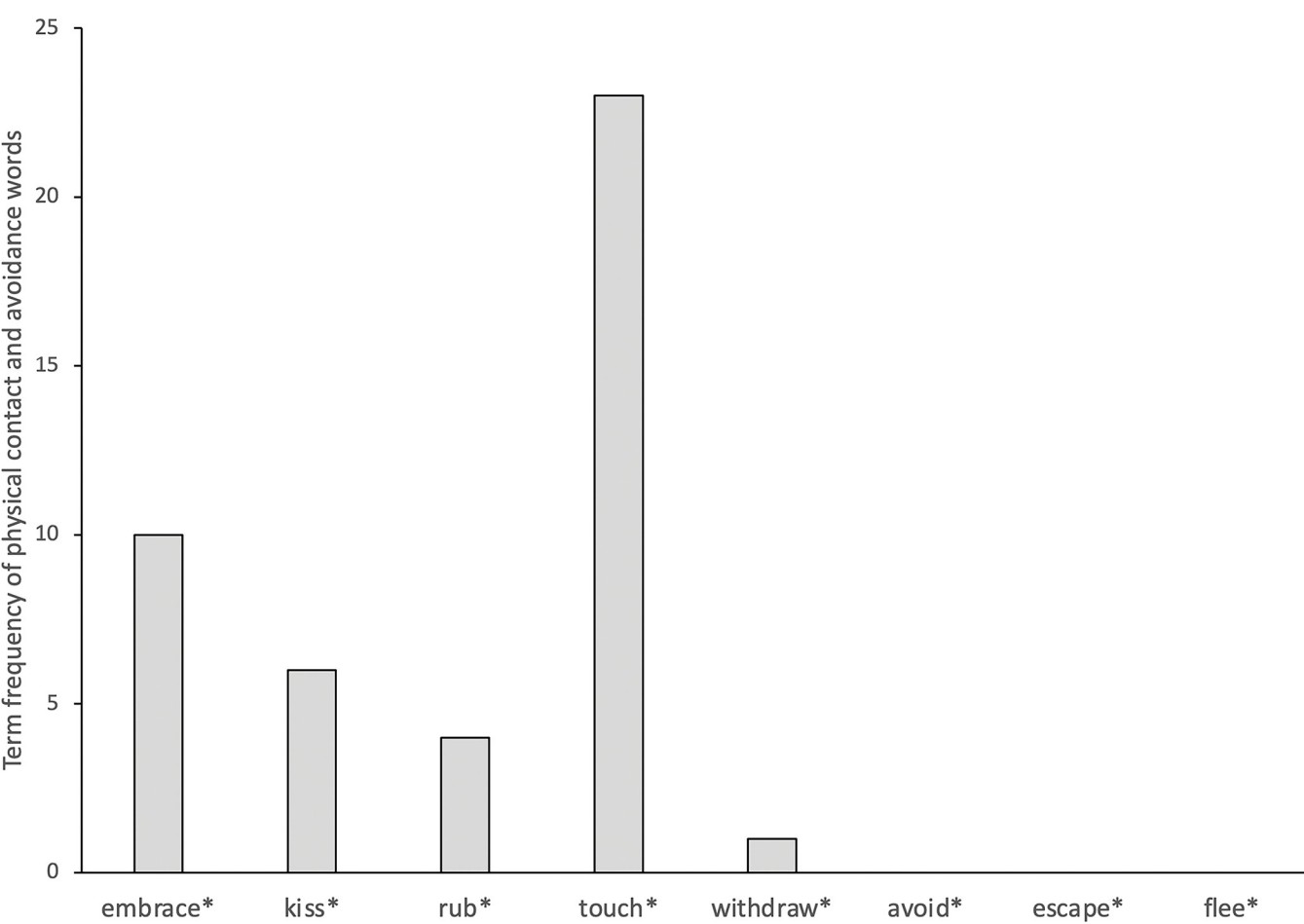

**Fig 1. Frequencies of physical contact and avoidance words in the context of explorer-indigenous people interactions.** Asterisks next to words indicate that conjugations of each given verb were also included in the analysis.

manifest avoidance of physical contact. For instance, the word "disgust" was used only twice, never to describe the native people. There was only one occurrence of the word "repulsive", which was describing same-sex unions among the men in a particular Native American tribe. Other synonyms, or near-synonyms of disgust (e.g., repugnant, repulsive, repugnance, revolting, revolted) were not present in the corpus. In contrast, the explorers described the bodies and morphological characteristics of indigenous people as pleasant, handsome and beautiful. There were many occurrences of physical contacts between the explorers and indigenous people. The interaction between Portuguese explorer Cadamosto and Guineans is a case in point. Cadamosto wrote that Guineans were surprised by his skin color and touched his body many times, rubbing his skin with their spittle to see if his white skin would peel off. Neither Guineans avoided touching an individual with unfamiliar skin color nor did Cadamosto report even a hint of repugnance of being touched with the spittle of Guineans. Italian explorer Christopher Columbus also reported several instances of physical contact that involved indigenous people kissing the hands and feet of the explorers, holding their arms, and putting their hands on their heads. French explorer Jacques Cartier's narrative also indicates that both adults and children made frequent physical contacts with European explorers and brought blind and sick individuals to have the explorers touch their ailing body parts to cure them. Cartier recorded

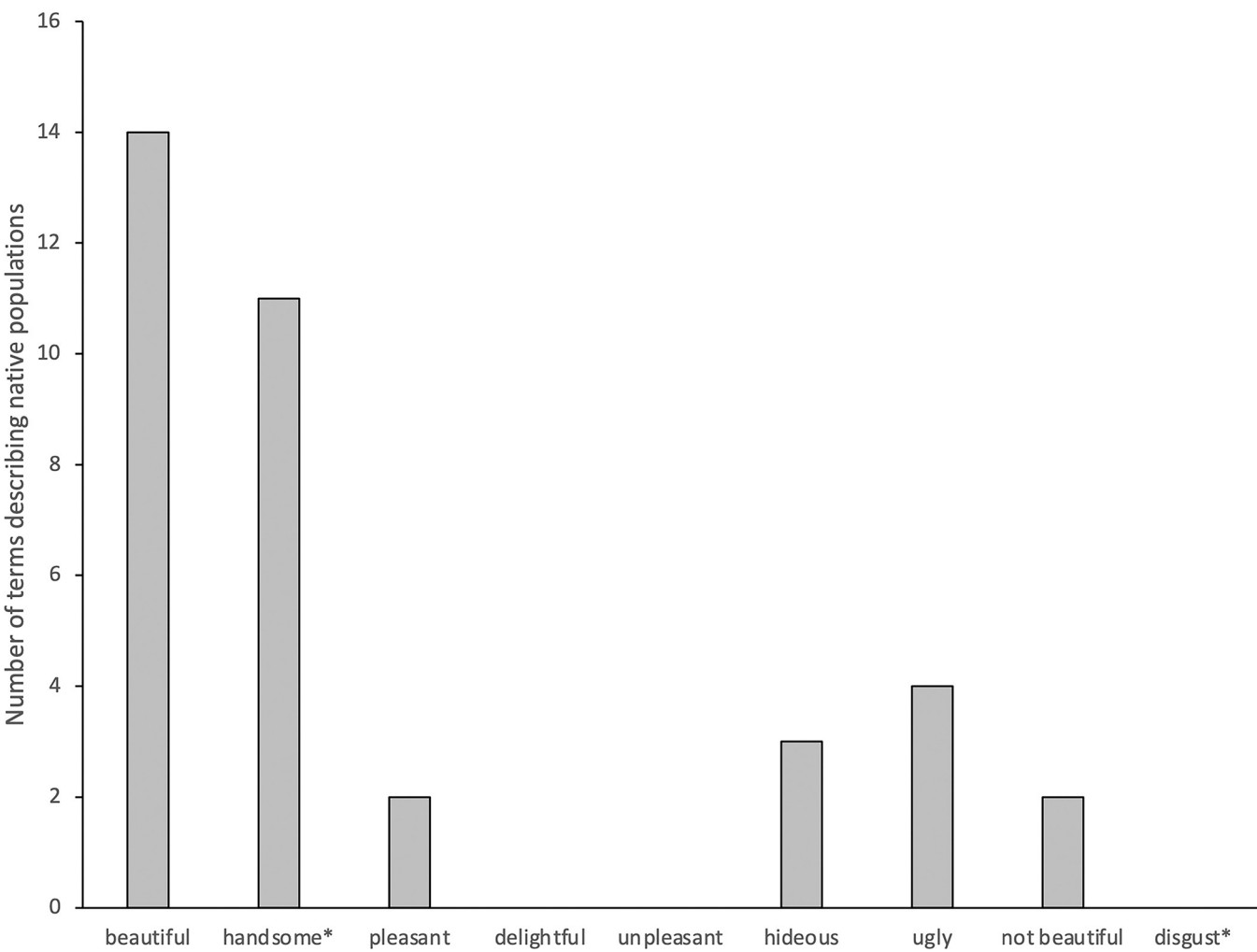

**Fig 2. Frequencies of words explorers used in their description of indigenous people.** Asterisks next to words indicate that conjugations of each given verb were also included in the analysis.

many instances of physical contacts such as stroking the faces and other body parts of the explorers. Spanish explorer Cabeza De Vaca reported similar observations. Rather than perceiving the European explorers as disease risk, the natives perceived them as healers and asked them many times to sign on their sick with the cross to heal them. The natives touched the explorers' faces and bodies and then touched their own hands and bodies. There are several instances of physical contact in Italian explorer Verrazano's account as well. For instance, a crew member who fell to the ocean and swam to the land for safety was rescued by the native people. Verrazano wrote that the natives seemed surprised by the unfamiliar look of the European man, and then they touched him several times and hugged him with affection. Cabral reported similar incidents of physical contact, such as being touched and embraced by the indigenous people.

Overall, our reading of the interactions between the indigenous people and European explorers between the 15th and 16th centuries did not reveal a pattern that would support the disease avoidance theory of xenophobia.

## Study 2

The small size of the corpus in our study may have been a limiting factor in drawing conclusions to test the xenophobia prediction of the disease avoidance theory. With the invention of steamships in the 19th century, travel became more affordable and voyages to distant locations were shortened, leading to a growing number of Europeans travelling for leisure and writing about their experiences in exotic locations, especially in Africa. A growing number of travelers, including government officers, hunters, adventurers, and missionaries, published hundreds of books in various forms such as diaries, reports, daily logs in large quantities during the 19[th] century.

We compiled a corpus of such travel narratives published between 1795 and 1930, and analyzed it to test the disease avoidance theory of xenophobia. However, reading a massive number of texts and considering multiple factors to evaluate a theory in an unbiased manner exceeds human capacities [27]. In recent decades, computational natural language processing models such as word embedding algorithms have emerged to extract statistical regularities from massive amounts of texts. Word embedding algorithms are based on the distributional hypothesis in semantic theory, according to which, words that tend to cooccur share similar contextual meanings [28]. These algorithms model words in large corpora, where each word is coded by a vector in a multidimensional space. Consequently, words used in the same semantic contexts are localized closer to each other, implying a semantic similarity between them. Recent studies utilizing *the d score* [13, 29, 30] have shown that word embedding bias can track implicit associations in textual corpora measured with word embedding association test (WEAT), revealing implicit gender biases and ethnic stereotypes, and widely shared cultural associations [31]. We used the d score (see Appendix B in S1 File) to determine whether the 19[th] and early 20[th] century travel literature shows an association between descriptions of unfamiliar indigenous people and words connoting disgust, and hence, disease avoidance.

## Method

We used the Global Vectors for Word Representation (GloVe) algorithm [32], which creates a cooccurrence probabilities matrix between words in a corpus by first transforming them to one-hot vectors and then using an unsupervised neural network algorithm to learn their contextual similarities by calculating the cosine similarities between the vectors before categorizing them based on the shared contextual similarities in a multidimensional semantic space.

**Selection of resources.**   We formed a corpus composed of travel narratives published between 1795 and 1935 by using a list of keywords (see Appendix C in S1 File) to search for these texts. We selected texts written in English without subject restrictions by authors who had first-hand experience in the Americas, Africa and Australia. Our search on gutenberg.com and archive.org yielded 466 texts (The list of texts constituting the corpus is provided in the Open Science Framework repository, https://osf.io/4f7ty/). We transformed the texts to lower case letters, removed all punctuation, digits, and symbols. Then, we tokenized the corpus using the Stanford Natural Language Package after merging the texts. The corpus size was 255 MB with approximately 47 million tokens. Although a large corpus size is desirable for generating stable embeddings, the GloVe algorithm has an advantage in that it can produce stable embeddings even with relatively small corpus sizes [32, 33]. Simulation studies have confirmed that when there is a specific focus, domain specificity is more critical than corpus size for generating meaningful embeddings [34–36].

**Hyperparameters**: The algorithm scanned the corpus with 15 iterations with a symmetric window size of 10 words to identify the words that cooccurred within this window. We set the vector dimension to 300 and the minimum vocabulary count to 10 (The corpus, and all of the associated files are provided in the Open Science Framework repository, https://osf.io/4f7ty/).

**Selection of the context words**: We generated two sets of context words from the corpus by following the method we describe in Appendix D in S1 File, which involved identifying synonymous words using the Merriam Webster dictionary and selecting words with relatively strong cosine similarity scores with each other. We also chose words with weaker cosine similarity scores from the other set, and each context word had a term frequency of at least 100 in the corpus. Consistent with the positivity bias observed in textual corpora [37], the mean frequency of the pleasant-laden words ($M_{pleasant}$ = 5169) was higher than that of the disgust-laden words ($M_{disgust}$ = 442). Internal consistency of the disgust-laden words was very good with Cronbach alpha values ranging from 0.89 and 0.96 throughout the analyses.

**Selection of the group names**: We calculated the term frequencies of the group names mentioned in the narratives and those that appeared in a relatively large number of narratives to avoid reflecting the biases of a few authors (see Appendix D in S1 File for the frequency of group names across the corpus). Next, we generated two lists of group names, one for native groups and the other for non-native groups, using the method described above. A permutation test showed that both sets of group names had similar frequencies ($M_{native}$ = 4856, $M_{nonnative}$ = 4628), p = 0.91.

To test the disease avoidance account of xenophobia, we calculated embedding bias indexed by the *d score* [13, 29, 30]. Due to the relatively small number of target words (N = 13 in each set), we used Hedge's adjusted g formula to control for positive bias in the calculation of effect sizes [38]. We first calculated embedding bias between the native and non-native group names based on their differential association with the context words. We also calculated embedding bias of morphological characteristics to determine if they were embedded with disgust more than pleasantness. Finally, we used network analysis method to discover the characteristics of the native group names associated with the context words.

## Results

We validated our list of context words between the names of various edible fruits and the disease-causing objects (see Appendix D in S1 File). A permutation test confirmed the existence of an embedding bias, $g_{adj}$ = 3.57, p < 0.001 (Fig 3, left panel). Specifically, the disease-causing objects were embedded with disgust-laden words more than pleasant-laden words. This trend was reversed for the fruits. Then, we examined whether there was an embedding bias for the native group names similar to the disease-causing objects compared to the non-native group names. An identical analysis confirmed that this was the case, $g_{adj}$ = 1.66, p < 0.001 (see Fig 3, middle panel). Next, we tested whether words related morphological characteristics (see Appendix D in S1 File) were more associated with the context words of disgust than the context words of pleasantness as the disease avoidance theory would anticipate. However, a permutation test did not show a significant difference, p = 0.40 (Fig 3, right panel). Although the frequency of words is a crucial parameter that can affect the formation of word embeddings [39, 40], it fails to explain the differential association between a specific set of context words and a particular category of group names given the group names (also fruit vs. disease agents) exhibit statistically similar word frequencies (see Appendix D in S1 File).

As we did not find embedding bias between the morphological characteristics and disgust-laden context words, we sought to identify the specific characteristics of the native groups, rather than their names, that may have been embedded with the context words. It is possible that certain words in the corpus mediated the relationship between the group names and the context words, which may not have been easily detectable solely through word embedding analysis. To explore this possibility, we employed network analysis equipped with graph, as suggested by [41], to uncover latent semantic relationships between words. Using Gephi

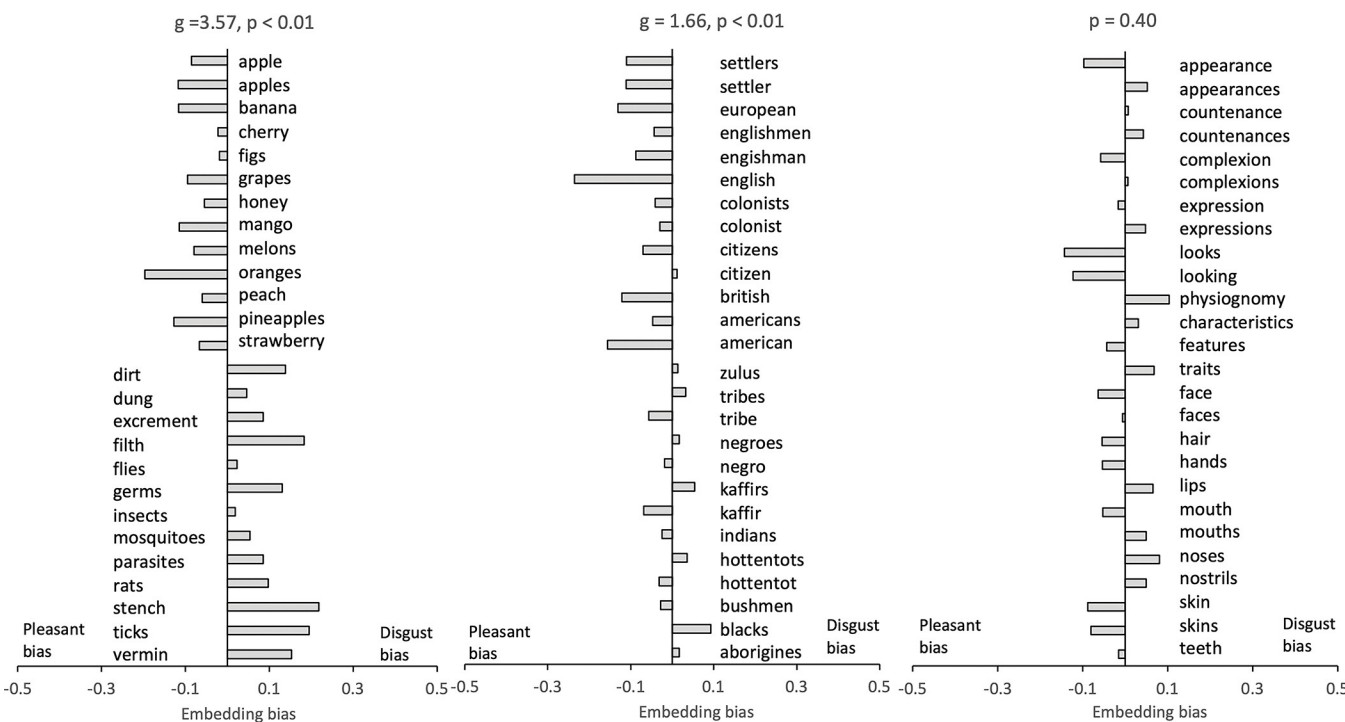

**Fig 3. Embedding bias of the two sets of context words with various target words.** The x axis shows the d score. Left panel: Embedding bias of the two sets of context words with disease-causing object names and fruit names. Middle panel: Embedding bias of the two sets of context words with native and nonnative groups. Right panel: Embedding bias of the two sets of context words with facial morphological characteristics.

software [42], we constructed a network that included the native and non-native group names, both sets of context words, and the words embedded with them. Nodes and edges represented words and their cosine similarities, respectively. We applied the forced atlas algorithm, which positions nodes to reduce the number of edge crossings by using forces of attraction and repulsion between nodes [43], and sliced the network at 0.20 cosine similarity score. The algorithm positioned nodes closer based on their cosine similarities and the number of nodes (e.g., embedded words) they share. For clarity, we present a simplified version of the network in Fig 4A. To identify the embedded words mediating the association between the group names and the context words, we calculated the betweenness centrality index [44] after filtering nodes with smaller degrees (k < 3). We found that many of the words demonstrating strong betweenness centrality were the terms such as "savages" that frequently used by European travelers in depicting native groups, (Fig 4B). We then conducted a multiple regression analysis to examine the relationship between the words in each set and the betweenness centrality coefficient scores of words larger than 0.001. The results revealed that the set of native group names was a significant predictor of the betweenness centrality scores, HHHÓÓÔÔ₁₁₁₁ ∫∫∫(ß = 0.008, t (36) = 2.99, p = 0.005 as was the set of disgust-laden words, ß = 0.007, t (36) = 2.69, p = 0.01. Similar analyses did not reveal significant association between the betweenness centrality scores and the cosine similarity scores of the words in non-native groups words or the pleasant-laden words, the largest t (36) = -1.16, p = 0.25. We also calculated the embedding bias of the centrality words to determine how much they are associated with the context words and the group names. Permutation tests revealed that these words were associated with the native groups, $g_{adj}$ = 1.44, p < 0.001 and the disgust-laden context words, $g_{adj}$ = 1.26, p = 0.007, (see Fig 5).

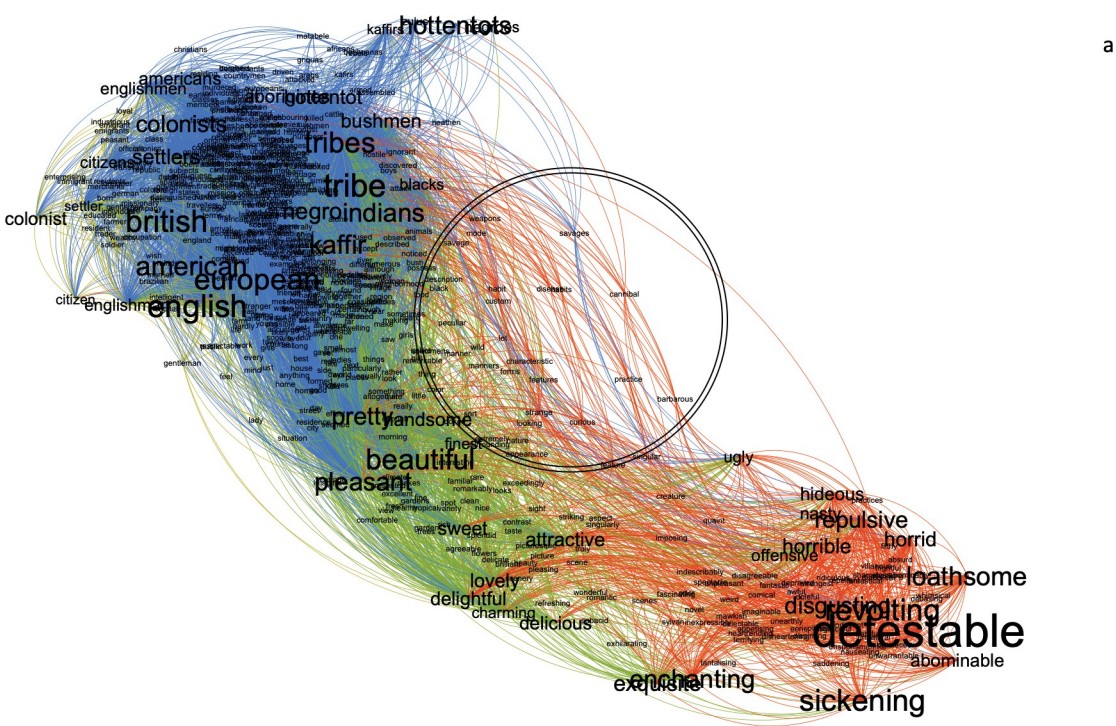

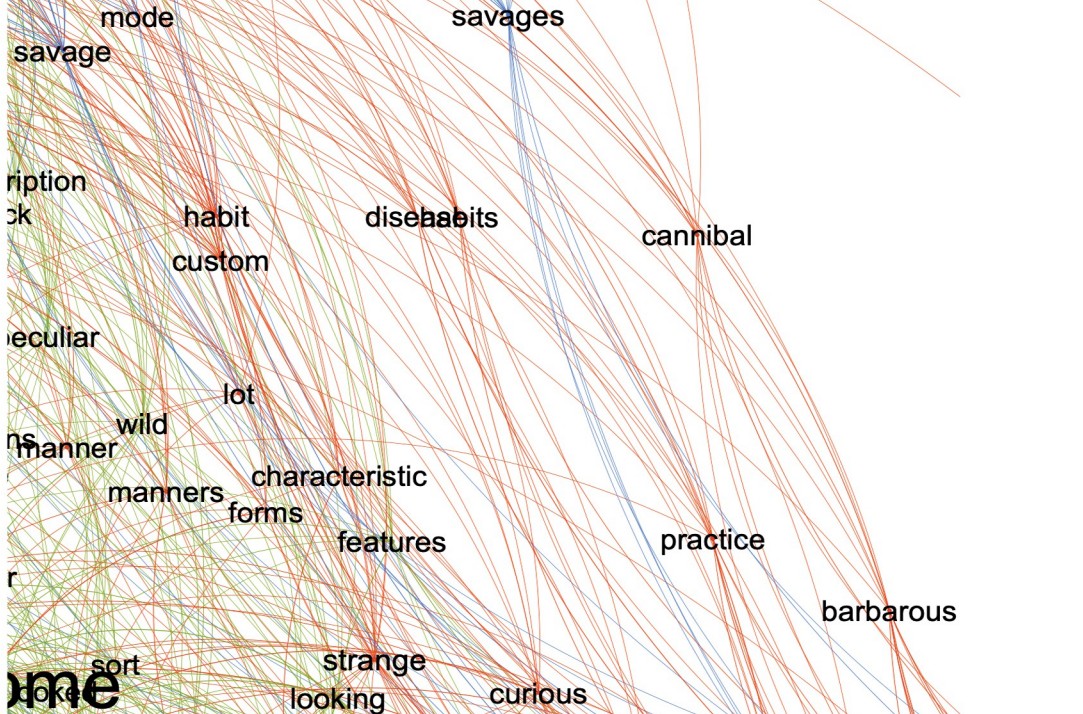

**Fig 4. a**. The semantic network of the group names, the context words and their embedded words. The network was sliced at minimum 0.20 cosine similarity since the mean cosine similarity between group names as well as between the context words generally ranged between 0.20 and 0.50. Nodes with degrees smaller than eleven edges were filtered out for the sake of simplicity and visibility. The network was subjected to a community detection algorithm [45] to present the words that coalesced together and the edges connecting the nodes, for visual clarity with a resolution parameter of 1.0. The section of the network with the highest betweenness centrality are illustrated within a circle. **b**. The enlarged section of the network showing the nodes with the highest betweenness centrality index coefficients.

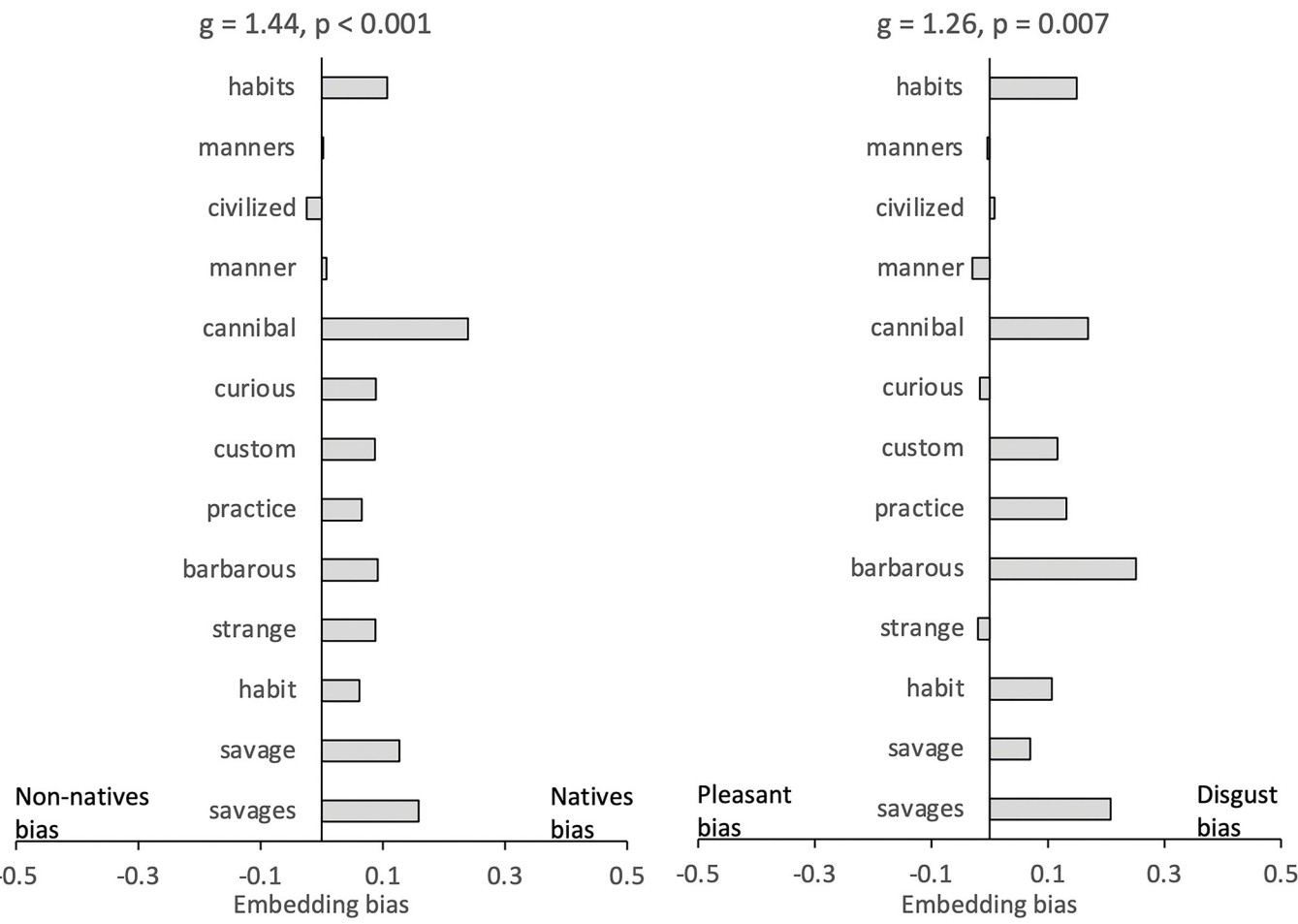

**Fig 5. Thirteen words with the highest betweenness centrality coefficient in order, and their differential embeddings with the native group names and the disgust-laden context words versus the nonnative group names and the pleasant-laden context words.**

## Discussion

The results indicated an embedding bias in the association between native groups and disgust-laden words. Importantly, although some Western travelers expressed repugnance toward the morphological characteristics of Africans, we did not find statistical evidence to support the anticipated bias suggested by the disease avoidance theory of xenophobia [21, 24]. Our network analysis revealed that the embedding bias for the native groups and disgust was mediated by words connoting inferiority, such as being "savage" and cultural characteristics such as "habits" and "customs". It appears that the association between disgust and xenophobia, which was widely reported in colonial travel literature, is mediated by the perception of the inferior status of native people and their cultural differences, rather than by their unfamiliar appearance.

## General discussion

History as an immense database of various behaviors, thoughts, and emotions expressed and recorded by diverse individuals in different settings can be mined and analyzed to extract regularities to test social psychological theories. Certain unique historical events, such as the first

encounters of Europeans with previously unknown peoples, may contain crucial information that cannot be obtained through any other research method. Despite this, psychologists have not yet paid sufficient attention to study such rare historical occurrences. Moreover, historical data can be used to assess the transhistorical validity of many social psychological theories, particularly the extent to which evolutionary theories in social psychology are temporally invariant.

One such theory, the behavioral immune system theory [21, 24] explains prejudice and xenophobia with the exaptation of the disease avoidance adaptation posited to have been acquired during evolution. The existence of this adaptation is well-supported by evidence from many animal species [46]. Our hypothesis was that if disease avoidance is the underlying mechanism for xenophobia, then disease and foreigner association should be a common and persistent theme throughout history. While some scholars have presented historical evidence to support the idea of disease avoidance playing a role in prejudice and xenophobia [24, 25] the evidence they provide is anecdotal and thus may be influenced by biased selection rather than objective analysis of a large historical textual corpus.

The word frequency analysis of the corpus that we compiled from the first encounters between European explorers and native populations in the 15th and 16th centuries did not reveal an association between unfamiliarity and disease avoidance. Despite the acute presence of disease threats, there were many physical contacts between the explorers and native people without the slightest hint of repugnance. It should be noted that in many cases, these contact behaviors were often initiated by the native people who believed that the European explorers descended from the heavens and would be able to cure their maladies. In contrast to using disgust-laden words, Europeans often described indigenous people as beautiful, and handsome also found their appearance and habitations to be clean and healthy.

We note that drawing conclusions from mere statistical analyses of historical corpora without considering political, religious, economic, and other societal factors that might have influenced historiography is fraught with problems. For instance, European explorers, may have held positive biases toward native populations, portraying them as submissive and open to religious conversion. This portrayal could have been influenced by their pursuit of administrative privileges and financial support from the European monarchs they served [47, 48]. However, given the high frequency of physical contact behaviors, often initiated by the natives, the evidence still contradicts the disease avoidance theory of xenophobia. Furthermore, one must also consider the personalities of the explorers themselves. It is plausible that European explorers shared a strong sensation-seeking trait. This possibility may reduce the generalizability of the findings we can draw from the corpus.

Historical data extracted from the travel literature of the 19th and early 20th centuries suggested the existence of an association between disgust and xenophobia. However, this association did not appear to be a consequence of the unfamiliarity with native people. Instead, our analysis showed that embeddings between the native group names and the disease avoidance words were mediated by words connoting perceived inferiority. The majority of travel narratives from this era indeed described native people as savages and portrayed them and their living conditions with repugnance [49–51].

Word embedding analyses have inherent limitations and potential pitfalls. One of these limitations is the corpus size, which can affect the stability of embeddings. Unfortunately, it is challenging to avoid this limitation when working with historical corpora. However, the GloVe algorithm has demonstrated the ability to detect cosine similarities between word vectors even with relatively small corpora. Another potential pitfall is the possibility of embedding bias due to frequency difference between sets of attributes or target words. Studies have revealed the existence of a positivity bias in many texts, where authors tend to use more

positive words than negative ones [52]. However, we found that this bias did not explain the embedding bias we identified.

In conclusion, it appears that the association between xenophobia and disgust has been shaped by the cultural changes that took place throughout history. It is important to note that while evolutionary psychology highlights the role of natural selection in shaping human behavior, it does not claim that all aspects of behavior and cognition are solely determined by evolved adaptations. It recognizes the influence of cultural factors and acknowledges the complex interplay between genes, environment and behavior. As Rozin [53, 54] explains while disgust is a universal emotion, it is highly susceptible to cultural learning and transmission. Therefore, without transhistorical or cross- cultural analysis, it is unfeasible to ascertain the degree to which an observed psychological phenomenon is attributed to an evolved adaptation or a cultural adaptation. Consequently, as Henrich, Heine and Norenzayan argue [55], what researchers investigate may not necessarily represent psychological universals but instead adaptations that are specific to particular cultures, which are mistakenly presumed to be psychological universals.

## Supporting information

**S1 File.**
(DOCX)

## Author Contributions

**Conceptualization:** Ceyhun Sunsay.

**Data curation:** Ceyhun Sunsay.

**Formal analysis:** Ceyhun Sunsay.

**Investigation:** Ceyhun Sunsay.

**Methodology:** Ceyhun Sunsay.

**Project administration:** Ceyhun Sunsay.

**Resources:** Ceyhun Sunsay.

**Software:** Ceyhun Sunsay.

**Supervision:** Ceyhun Sunsay.

**Validation:** Ceyhun Sunsay.

**Visualization:** Ceyhun Sunsay.

**Writing – original draft:** Ceyhun Sunsay.

**Writing – review & editing:** Ceyhun Sunsay.

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
