## [Decision Letter · Decision Letter 0]

26 Jun 2023

PONE-D-23-15538A Historical Evaluation of the Disease Avoidance Theory of XenophobiaPLOS ONE

Dear Dr. Sunsay,

Thank you for submitting your manuscript to PLOS ONE. After careful consideration, we feel that it has merit but does not fully meet PLOS ONE’s publication criteria as it currently stands. Therefore, we invite you to submit a revised version of the manuscript that addresses the points raised during the review process.

We look forward to receiving your revised manuscript.

Kind regards,

Grant Rich, Ph.D.

Academic Editor

PLOS ONE

Journal Requirements:

" The funders had no role in study design, data collection and analysis decision to publish, or preparation of the manuscript."

4. Please upload a new copy of Figure 4 as the detail is not clear. Please follow the link for more information: " ext-link-type="uri" xlink:type="simple">https://blogs.plos.org/plos/2019/06/looking-good-tips-for-creating-your-plos-figures-graphics/"
" ext-link-type="uri" xlink:type="simple">https://blogs.plos.org/plos/2019/06/looking-good-tips-for-creating-your-plos-figures-graphics/"

Additional Editor Comments:

Dear Author, two peer reviewers and I have now carefully reviewed your submission, and we all agree the decision is "minor revision"

Please address the comments by both reviewers, including making more reference to existing historiometric research in psychology such as by Dean Keith Simonton, and clarify the conclusions/theory and data analysis as especially noted by reviewer 2

Dr Rich

Juneau, Alaska

REVIEWER ONE WROTE MINOR REVISION

A very well-conceived study that makes an incremental contribution to an important substantive question. I only have two recommendations for improvement, one more major than the other.

In the more minor category, although the author specifies two studies, only one heading is provided, namely that for Study 2. Presumably a heading for Study 1 should appear somewhere before the first Method section.

On the more major side, the author apparently believes that psychologists have not taken advantage of historical data. That may be true for this very specific substantive question, but the fact remains that psychologists have been using historical data for well over a century. Indeed, such usage occurred prior to laboratory experiments! Moreover, the data goe back not just centuries but sometimes millennia. For general reviews, see

Simonton, D. K. (2003). Qualitative and quantitative analyses of historical data. Annual Review of Psychology, 54, 617-640.

Simonton, D. K. (2009). Historiometry in personality and social psychology. Social and Personality Psychology Compass, 3, 49-63.

Hence, the author should make it clear that they are talking about a very narrow substantive issue that has failed to take advantage of this resource.

REVIEWER TWO MINOR REVISION

The article is interesting as it examines the writings of European explorers to see if there is any support for the disease avoidance theory of xenophobia. It is well written. Although it is easy to understand the basis of the study, its discussion, and conclusions, I found the statistical and mathematical analysis difficult to understand—these aspects should be evaluated by reviewers familiar with these types of analyses.

I had a couple of small concerns when I read the paper. The document has no pagination, so I will reference my comments with quotes from the text as needed.

Results Section: The author conducted chi-squared tests based on data reported in Figure 1. It would be helpful to explain where the 45 and 36 degrees of Freedom come from and what difference is being tested. In Figure 1 there are only 5 words depicted and their frequencies. Thus, it is not easy to follow what is being tested.

Figure 3 is hardly readable where there is an overlap between words and the horizontal bars. Also, Figure 5 is not easy to read even when you enlarge it.

“occurrence of the word “repulsive”, but it was describing same-sex unions among Native…”

Comment: The word “but” in the above sentence needs clarification, or just remove the word, so as not to sound judgmental. It is not clear if the context of other words was considered in the analyses.

“Overall, our reading of the interactions between the indigenous people and European explorers between the 15th and 16th centuries did not reveal a pattern that would support the disease avoidance theory of xenophobia.”

Comment: It is not clear if this is based on “word counts” in the writings of European explorers or a content analysis of the writings or both.

It is impressive that the analyses did not support the disease theory of prejudice per these writings—but is it possible that the writings may have been biased on the part of explorers who minimized the use of such words so as not to appear biased? Is there a possibility of any political reasons for writing that the natives were welcoming of them, and hero-worshipped them as sent from heaven? On the other hand, the explorers were adventurers and may have had a more open attitude toward the natives and showed interest in them when they interacted with th

Reviewers' comments:

Reviewer's Responses to Questions

**Comments to the Author**

1. Is the manuscript technically sound, and do the data support the conclusions?

Reviewer #1: Yes

Reviewer #2: Yes

2. Has the statistical analysis been performed appropriately and rigorously? 

Reviewer #1: Yes

Reviewer #2: I Don't Know

3. Have the authors made all data underlying the findings in their manuscript fully available?

Reviewer #1: Yes

Reviewer #2: Yes

4. Is the manuscript presented in an intelligible fashion and written in standard English?

Reviewer #1: Yes

Reviewer #2: Yes

5. Review Comments to the Author

Reviewer #1: A very well-conceived study that makes an incremental contribution to an important substantive question. I only have two recommendations for improvement, one more major than the other.

In the more minor category, although the author specifies two studies, only one heading is provided, namely that for Study 2. Presumably a heading for Study 1 should appear somewhere before the first Method section.

On the more major side, the author apparently believes that psychologists have not taken advantage of historical data. That may be true for this very specific substantive question, but the fact remains that psychologists have been using historical data for well over a century. Indeed, such usage occurred prior to laboratory experiments! Moreover, the data goe back not just centuries but sometimes millennia. For general reviews, see

Simonton, D. K. (2003). Qualitative and quantitative analyses of historical data. Annual Review of Psychology, 54, 617-640.

Simonton, D. K. (2009). Historiometry in personality and social psychology. Social and Personality Psychology Compass, 3, 49-63.

Hence, the author should make it clear that they are talking about a very narrow substantive issue that has failed to take advantage of this resource.

Reviewer #2: The article is interesting as it examines the writings of European explorers to see if there is any support for the disease avoidance theory of xenophobia. It is well written. Although it is easy to understand the basis of the study, its discussion, and conclusions, I found the statistical and mathematical analysis difficult to understand—these aspects should be evaluated by reviewers familiar with these types of analyses.

I had a couple of small concerns when I read the paper. The document has no pagination, so I will reference my comments with quotes from the text as needed.

Results Section: The author conducted chi-squared tests based on data reported in Figure 1. It would be helpful to explain where the 45 and 36 degrees of Freedom come from and what difference is being tested. In Figure 1 there are only 5 words depicted and their frequencies. Thus, it is not easy to follow what is being tested.

Figure 3 is hardly readable where there is an overlap between words and the horizontal bars. Also, Figure 5 is not easy to read even when you enlarge it.

“occurrence of the word “repulsive”, but it was describing same-sex unions among Native…”

Comment: The word “but” in the above sentence needs clarification, or just remove the word, so as not to sound judgmental. It is not clear if the context of other words was considered in the analyses.

“Overall, our reading of the interactions between the indigenous people and European explorers between the 15th and 16th centuries did not reveal a pattern that would support the disease avoidance theory of xenophobia.”

Comment: It is not clear if this is based on “word counts” in the writings of European explorers or a content analysis of the writings or both.

It is impressive that the analyses did not support the disease theory of prejudice per these writings—but is it possible that the writings may have been biased on the part of explorers who minimized the use of such words so as not to appear biased? Is there a possibility of any political reasons for writing that the natives were welcoming of them, and hero-worshipped them as sent from heaven? On the other hand, the explorers were adventurers and may have had a more open attitude toward the natives and showed interest in them when they interacted with them.

6. PLOS authors have the option to publish the peer review history of their article (what does this mean?). If published, this will include your full peer review and any attached files.

Reviewer #1: No

Reviewer #2: No

Grant J. Rich, PhD 

Candidate for President-Elect for the American Psychological Association

President-Elect Society for Peace, Conflict, and Violence (APA)

President-Elect Society for Media Psychology and Technology (APA)

Fellow, Association for Psychological Science (APS)

Fellow, American Psychological Association (APA) (D1, D2, D46, D48, D52)

Senior Contributing Faculty, Walden University

Editorial Board Member: PLOS ONE, APA's Peace Conflict,
APA's Traumatology

Book Series Co-Editor w/ Anthony Marsella (U. Hawai'i), Springer. International and Cultural Psychology (ICUP)       

 https://www.springer.com/series/6089

Select Recent Books

(Rich, Gielen, Takooshian,
2017).
Internationalizing the Teaching of Psychology.

IAP.

(Rich Sirikantraporn, 2018).
Human Strengths and Resilience: Cross Cultural and International Perspectives. Rowman Littlefield.

(Rich, Jaafar, Barron, 2020).
Psychology in Southeast Asia. Routledge.

(Rich Ramkumar, 2022).
Psychology in Oceania and the Caribbean. Springer. 

(Rich, Kuriansky, Gielen, Kaplan, 2023)in press .
Psychosocial Experiences and Adjustment of Migrants: Coming to the USA. 
Elsevier.

(Rich, Kumar, Farley, in contract).
Handbook of Media Psychology and Technology-The Science and the Practice. Springer. 

---

## [Author Response · Author response to Decision Letter 0]

6 Sep 2023

Reviewer 1

 The reviewer is absolutely correct in their comment regarding the long history of using historical data. As the reviewer indicates, calls for the use of historical data go back to the earlier years of psychology, especially in France (e.g., Meyerson). The use of historical data spanning several centuries with an explicit objective to test the historical validity of evolutionary theories is the novel idea in this study. This is especially true in the context of the first encounters between Europeans and newly discovered non-European populations and the disease avoidance theory of xenophobia. Nonetheless, I agree with the reviewer for their invitation to credit Simonton’s studies. Although the objective of using historical data was in a different line of research, Simonton is indeed a pioneer in using historical data in contemporary psychology. It is an oversight on my part to not cite his studies despite an earlier email communication with him. I revised the manuscript accordingly clarify that the manuscript here uses historical data to test the historical validity of the disease avoidance theory of xenophobia. 

 I added heading for Study 1 as the reviewer suggested. 

Reviewer 2

1. I agree that presentation of the statistical analysis in study 1 was not clear. I have added a sentence to the method section and revised the results section to clarify which data were analyzed. Specifically, I rephrased the paragraph to explain that the degrees of freedom represent the total number of combined contact and contact avoidance behaviors, and the Chi Square analysis tested whether these words equally distributed between these two categories. The same approach was applied to combined disease avoidance words and their antonyms. Furthermore, I have adjusted the figure captions of figure 1 to clarify that asterisks next to each word indicate that conjugations of target words were also included in the analyses (e.g., disgust, disgusted, disgusting etc.) 

2. To enhance clarity. I have removed the overlap between words and the horizontal bars in figure 3. In Figure 5, I conducted a modularity analysis enabling me to assign colors to nodes and edges, based on their respective communities. I increased the font size and presented the network and the section featuring words with the highest betweenness centrality index separately, making the figure more visible.

3. I revised the sentence in the discussion section, aiming to eliminate any judgmental tone. The intention behind the sentence was to emphasize that the term 'repulsive' did not pertain to the appearance of Native Americans. 

4. I have rephrased the sentence of this paragraph to clarify that the word counts analysis of the texts did not support the disease avoidance theory of xenophobia. Furthermore, I have agreed to include a paragraph addressing the question raised by the reviewer. Indeed, there were several political and religious motivations that may have led European explorers to portray native populations positively. While these possibilities may apply to the verbal descriptions of the natives, they do not account for the frequent physical contacts observed between the groups, particularly given that these contacts were initiated by the natives themselves. Nonetheless, the question raised by the reviewer is significant for historical interpretation. The common personality trait shared among the explorers is intriguing and warrants a discussion on the generalizability of our findings. I have attempted to address these concerns in the general discussion.

5. Page numbers were provided

---

## [Editor Report · Decision Letter 1]

9 Nov 2023

A Historical Evaluation of the Disease Avoidance Theory of Xenophobia

PONE-D-23-15538R1

Dear Dr. Sunsay

We’re pleased to inform you that your manuscript has been judged scientifically suitable for publication and will be formally accepted for publication once it meets all outstanding technical requirements.

Kind regards,

Grant Rich, Ph.D.

Academic Editor

PLOS ONE

Additional Editor Comments (optional):

Your article makes a valuable contribution- the authors have been responsive to the reviewers' comments
---

## [Editor Report · Acceptance letter]

5 Dec 2023

PONE-D-23-15538R1 

A Historical Evaluation of the Disease Avoidance Theory of Xenophobia 

Dear Dr. Sunsay:

I'm pleased to inform you that your manuscript has been deemed suitable for publication in PLOS ONE. Congratulations! Your manuscript is now with our production department. 

Kind regards, 

on behalf of

Dr. Grant Rich 

Academic Editor

PLOS ONE